# Epigenetic Factors as Etiological Agents, Diagnostic Markers, and Therapeutic Targets for Luminal Breast Cancer

**DOI:** 10.3390/biomedicines10040748

**Published:** 2022-03-23

**Authors:** Nguyen Xuan Thang, Seonho Yoo, Hyeonwoo La, Hyeonji Lee, Chanhyeok Park, Kyoung Sik Park, Kwonho Hong

**Affiliations:** 1Department of Stem Cell and Regenerative Biotechnology, Institute of Advanced Regenerative Science, Konkuk University, Seoul 05029, Korea; thang.nx1012@gmail.com (N.X.T.); upreference98@naver.com (S.Y.); hyunwoo1001@naver.com (H.L.); affogatojoa@gmail.com (H.L.); chpark0729@gmail.com (C.P.); 2Department of Surgery, School of Medicine, Konkuk University, Seoul 05029, Korea; 20090117@kuh.ac.kr

**Keywords:** epigenetic factors, luminal breast cancer, chromatin structure, gene expression

## Abstract

Luminal breast cancer, an etiologically heterogeneous disease, is characterized by high steroid hormone receptor activity and aberrant gene expression profiles. Endocrine therapy and chemotherapy are promising therapeutic approaches to mitigate breast cancer proliferation and recurrence. However, the treatment of therapy-resistant breast cancer is a major challenge. Recent studies on breast cancer etiology have revealed the critical roles of epigenetic factors in luminal breast cancer tumorigenesis and drug resistance. Tumorigenic epigenetic factor-induced aberrant chromatin dynamics dysregulate the onset of gene expression and consequently promote tumorigenesis and metastasis. Epigenetic dysregulation, a type of somatic mutation, is a high-risk factor for breast cancer progression and therapy resistance. Therefore, epigenetic modulators alone or in combination with other therapies are potential therapeutic agents for breast cancer. Several clinical trials have analyzed the therapeutic efficacy of potential epi-drugs for breast cancer and reported beneficial clinical outcomes, including inhibition of tumor cell adhesion and invasiveness and mitigation of endocrine therapy resistance. This review focuses on recent findings on the mechanisms of epigenetic factors in the progression of luminal breast cancer. Additionally, recent findings on the potential of epigenetic factors as diagnostic biomarkers and therapeutic targets for breast cancer are discussed.

## 1. Introduction

Globally, breast cancer is among the top five diseases causing mortality [1,2]. Breast cancer accounts for approximately 25% of new cancer cases with an estimated 2.3 million cases diagnosed annually in both male and female. Additionally, breast cancer accounts for approximately 17% of all cancer-related deaths in women [2]. Early stages of breast cancer can be effectively screened and diagnosed using various techniques, such as magnetic resonance imaging, ultrasonography, clinical breast examination, and three-dimensional (3D) mammography [3,4]. The American Joint Committee on Cancer provides reference resources for tumor classification, genetic screening, and key prognostic and predictive factors for breast cancer precursors [5]. Breast cancer classification based on the expression of hormone receptors or other factors enables the development of effective treatment and the mitigation of cancer-related mortality [1,2,6,7]. Estrogen receptor (ER)-alpha (ERα) is the most well-known genetic biomarker for breast cancer. The expression of ERα is upregulated in 75–80% of all breast cancer cases [8,9,10].

Based on the tumor size and Human epidermal growth factor receptor-2 (HER2/erbB-2) expression status, patients with ERα-positive breast cancer are initially considered for endocrine therapy to reduce the risk of malignancy and tumor recurrence after treatment [6]. The mechanism underlying endocrine therapy involves the inhibition of ER signaling by suppressing the cancer cell utilization of 17β-estradiol (E2). Hormone therapies include selective ER modulators, such as tamoxifen and toremifene that inhibit ERα activity and steroidal and non-steroidal aromatase inhibitors (AIs) that target the E2 biosynthesis-related enzyme aromatase [11,12]. Fulvestrant, a selective ER degrader, competitively binds to ERα and promotes ERα degradation [13]. Clinical studies have demonstrated that adjuvant endocrine treatments improve the 5-year and 10-year survival rates of patients with ER-positive tumors [14,15]. Although endocrine therapy can mitigate the progression of cancer, approximately one-third of patients exhibit tumor recurrence within 15 years of endocrine therapy. Some studies have indicated that endocrine therapy promotes cancer metastasis by modulating the epigenome [16,17,18,19].

Both genetic and epigenetic factors are etiological agents for breast cancer. Tumorigenesis is mediated through multiple steps involving genetic and epigenetic changes, which mediate tumor initiation, progression, and heterogeneity by disrupting the equilibrium between oncogenes and tumor suppressor genes [20,21,22]. Previous studies have demonstrated that aberrant epigenetic changes are inheritable and that they are involved in initiating premalignant processes of invasive breast cancer by activating oncogenes [23,24] or silencing tumor suppressors [25,26,27].

In the last three decades, the mechanistic understanding of breast cancer has rapidly advanced. However, the contribution of epigenetic changes to the pathogenesis of breast cancer has not been completely elucidated. Complex factors, such as DNA methylation, histone modifications, and non-coding RNAs promote epigenetic changes in breast cancer [21,28]. Dysfunction of epigenetic regulators or epimutations can impair the repair of damaged DNA or alter the access of transcription factors to target genes. ER+ breast cancer is reported to harbor aberrant epigenomic changes that are closely associated with the initiation of tumorigenesis and the progression to metastasis [29,30]. Aberrant epigenetic regulation also plays a critical role in the tumorigenesis of ER+ breast cancer by transducing epithelial–mesenchymal transition (EMT) signals, modulating DNA methylation and histone modifications, and regulating oncogene expression [29,30,31,32]. Dysregulation of epigenetic mechanisms may contribute to the formation of “cancer progenitor cells” and promote drug resistance and metastasis [32].

This review focuses on the role of epigenetic changes in ER signaling and regulating oncogene expression in ER+ breast cancer and metastasis. Additionally, recent studies examining the epigenetic and transcription factors as potential diagnostic biomarkers and therapeutic targets to develop personalized medicine for ER+ breast cancer are summarized.

## 2. Epigenetic Regulation in ER Signaling

ER signaling is critical for physiological mammary gland development and pathological tumorigenesis. The healthy breast tissue is reported to comprise < 20% ERα-positive cells [33,34]. Hence, enhanced ER activity and ER-positive cell number are high-risk factors for breast carcinogenesis.

ERα stimulated by E2 directly binds to chromatin and regulates target gene expression or alters cytoplasmic signal transduction in non-genomic pathways [35,36,37]. The indirect mechanism of ERα involves the regulation of target genes in ER+ breast cancer by activating the Wingless-Int1 (Wnt), phosphatidylinositol 3-kinase/protein kinase B (PI3K/AKT), and mitogen-activated protein kinase (MAPK) signaling pathways [38,39,40]. An increase in estrogen receptivity and ER+ cells is considered a high-risk factor for breast carcinogenesis [41,42].

Pioneer factors are reported to be critical for the ER signaling pathway (Figure 1). For example, FOXA1 and GATA3 are enriched at the ERα-binding sites. This suggests that the absence of pioneer factors may lead to decreased ERα occupancy at its binding site upon E2 stimulation [43,44]. The physical interactions between these transcription factors and AP-2γ can locally unwind DNA in the heterochromatin regions. Subsequently, the transcription factors directly bind to their targets on the DNA sequence [44,45,46,47]. FOXA1, which plays an important role in ER-dependent breast cancer, promotes the loading of ERα transcription factors to a subset of their binding sites [43,48]. Factors, such as steroid receptor family, p160 family, inositol-requiring protein 80 (INO80), and pioneer factors directly interact with ERα to form coregulator complexes and recruit other transcription factors and activating factors to gene promoters [49,50]. Conversely, chromatin remodelers may recruit pioneer factors to target sites. Thus, these factors increase the accessibility of the chromatin structures [44,45,51]. Stimulation with E2 promotes the decondensation of nucleosomes through the chromatin remodeling complex of the SNF2 helicase superfamily members, including switching/sucrose non-fermenting (SWI/SNF), imitation switch, chromodomain helicase (CHD), and INO80 complex (Figure 1). These factors facilitate the recruitment of chromatin remodeling modulators and several helix–loop–helix ERα transcriptional coregulators and corepressors [50,52,53].

As shown in the Figure 1, E2 stimulation promotes the recruitment of P300, a histone H3 lysine 27 (H3K27) acetyltransferase, to the ERα-binding sites. Consequently, genes containing estrogen response elements (EREs) and ERE-like promoters are activated [52,53,54]. Estrogen signaling is also regulated by SNF2 family proteins, including the SWI/SNF and INO80 complexes, which are primarily involved in the modification of histones and the regulation of ERE-containing gene expression [50,55]. Histone acetyltransferases (HATs) may function as initiation signals in ER-mediated gene transcription in breast cancer [56]. ERα transcriptional coactivators recruit HATs to induce the expression of ERE-containing genes and alter the chromatin structure by acetylating H3K14, histone H4 lysine 5 and 8 (H4K5, and H4K8), and modulating H2A/H2B modification [57,58,59]. DNA demethylation is downregulated after the binding of pioneer factors to their targets that induce ERα responses and DNA repair in breast cancer [60]. Emerging evidence suggests that the methylation of histones is important for E2-responsive processes. For example, the depletion of the histone methyltransferase (HMT) enzyme lysine-specific demethylase 1, an enzyme catalyzing the demethylation of lysine 4 and 9 of histone H3 (H3K4 and H3K9), promotes E2-induced gene expression [61]. Furthermore, two polycomb repressive complexes (PRC), PRC1 and PRC2, are primarily involved in repressive chromatin function by catalyzing H3K27 methylation and H2A119 monoubiquitylation (H2A119ub1) at the enhancer and promoter regions of the ERα downstream genes during E2 induction [53,62,63,64]. H2Bub1 negatively regulates ER signaling by altering the enhancer valley of ERα-binding sites, repressing active histone marks for gene expression, and preventing transcription factor complex assembly to the sites [50,65,66]. SUMOylation of ERα mediated by SUMO1/sentrin/suppressor of Mif2-specific peptidase 2 (SENP2) promotes the recruitment of histone deacetylase 3 (HDAC3), which leads to the repression of estrogen-dependent and estrogen-independent proliferation of breast cancer cells [67,68].

## 3. Global Epigenome Profiles in ER+ Breast Cancer

Pioneer and transcription factors are frequently used to diagnose and classify breast cancer types as they are mutated or upregulated in ER+ breast cancer, especially in endocrine therapy-resistant metastatic breast cancer [69,70,71]. For example, approximately 20% of AI-resistant breast tumors exhibit mutations in the ERα-encoding gene, while approximately 15% of breast tumors exhibit resistance to anti-E2 therapy owing to downregulated ERα expression [72,73]. Previous studies have reported the mechanisms involved in inducing mutation in different genes, including those encoding FOXA1, ERα, or other ER coregulators, in ER+ breast cancer [74,75,76,77].

Advances in next-generation sequencing (NGS) technologies have revealed the critical roles of epigenetic factors in luminal breast cancer development and hormonal therapy resistance. Epigenetic dysregulation is frequently observed in the regulation of tumor suppressor genes and/or oncogenes in cancer cells [78]. Aberrant regulation of factors involved in ER signaling may promote breast cancer tumorigenesis. Thus, these factors are potential diagnostic biomarkers for breast cancer. Genomic instability is mediated by aberrant DNA methylation and histone modifications, which are associated with the prognosis of ER+ breast cancer subtypes [19,24,79]. Epigenetic factors are classified into writers, readers, and erasers and include DNA methyltransferases (DNMTs), ten-eleven translocation enzymes (TET), HATs, HDACs, HMTs, histone demethylases, and other factors, such as miRNAs, lncRNAs, and eRNAs [80,81,82]. Aberrant epigenetic functions are mechanically linked to epigenetic reprogramming during the formation of heterogeneous tumors and the adaptation to therapeutics. Some epigenetic factors involved in DNA methylation and histone modification are upregulated in ER+ breast cancer and are suggested to be diagnostic biomarkers for ER+ breast carcinogenesis, oncogenic reprogramming, and endocrine therapy resistance [69,83].

### 3.1. DNA Methylation Alterations in ER+ Breast Cancer

DNA hypomethylation or hypermethylation-mediated changes in epigenome and gene expression are associated with luminal breast cancer tumorigenesis and endocrine therapy resistance [84]. DNMTs and TETs regulate DNA methylation levels by catalyzing the addition or removal of covalent bonds between the methyl group and cytosine at CpG islands. DNA methylation status of the CpG islands at the promoter of tumor suppressor genes or oncogene regulators is closely related to changes in gene expression in cancer [85].

Various studies have suggested that global DNA methylation status can serve as a diagnostic biomarker for breast cancer [86]. A non-invasive diagnostic method can measure DNA methylation. In particular, circulating tumor DNA can be used to quantify DNA methylation and detect ERα-encoding gene mutations in patients with breast cancer [87,88]. Genome-wide DNA methylation profiling is a powerful approach that can comprehensively characterize the function of CpG islands in cancer recurrence and poor survival outcomes of patients with ER+ breast cancer [89]. DNA hypermethylation at the enhancer sites of ERE-containing genes regulates ER signaling. Conversely, ER signaling can induce DNA hypomethylation in ER-induced target genes [90,91]. Aberrant DNA methylation at the ERα-binding sites impairs the 3D chromatin interaction of topologically associating domains. The expression of ER-regulated genes is disrupted in endocrine therapy-resistant breast cancer [19]. Additionally, hypermethylation at the focal site in the promoter region of the ERα-encoding gene downregulates the ERα protein levels. This explains the signatures of approximately 40% of hormone therapy-resistant ER+ breast cancer cases [92,93,94].

DNA methylation can upregulate or downregulate *ESR1* expression in the in vitro and in vivo breast cancer models [93,95]. Various writers, readers, and erasers, which are aberrantly expressed in primary luminal breast cancer or metastasis, mediate DNA methylation. For example, the upregulation of DNMTs, such as DNMT1, DNMT3A, and DNMT3B, is correlated with endocrine therapy resistance in ER+ breast cancer [96,97]. Approximately 30% of all ER+ breast cancer cases exhibit upregulated expression of DNMT3B. Additionally, DNMT3B is co-expressed with DNMT1/3A in 3–5% of breast cancer samples [98]. Treatment with tamoxifen upregulates the expression levels of DNMT3A/B, whereas increased DNA methylation levels are accompanied by the upregulation of stem cell-like genes, including those encoding SRY-related HMG-box (SOX) and retinoblastoma (RB)-related family members, and the downregulation of *NRIP1*, *HECA*, and *FIS1* [99]. In contrast to TET1 and TET3, TET2 upregulation is frequently detected in ER+ breast cancer. Upregulated TET2 levels maintain the 5hmC levels and the components required for ERα-mediated transcriptional regulation at the ERα-binding sites [100]. TET2 binds to the active enhancers and promotes E2 response before the COMPASS and/or ERα/GATA3 complex to express ERE-containing genes [100,101]. The loss of TET2 is associated with the downregulation of *ESR1*, *GATA3*, and *FOXA1*, whereas treatment with 5-aza-2-deoxycytidine, a DNA hypomethylating agent, promotes ESR1 expression [95,102]. Similar to DNMT3A/B, TET2 is a potential biomarker for ER+ breast cancer progression and metastasis.

### 3.2. Aberrant Histone Modifications in ER+ Breast Cancer

#### 3.2.1. Histone Acetylation and Deacetylation

Previous studies have suggested that histone modifications can be utilized to identify breast cancer subtypes [103,104]. Furthermore, multiple evidence suggests that histone modifiers that elicit aberrant gene expression can be served as therapeutic targets for luminal breast cancer. Histone acetylation and deacetylation at the lysine (K) residues are reversible processes catalyzed by HATs and HDACs, respectively, in the presence of the acetyl-CoA cofactor. Dysregulation of HAT or HDAC functions is reported to be the characteristic signature of breast cancer subtypes and hormone therapy-resistant breast cancer [105,106,107,108]. HATs are categorized into the following three groups: the GCN5-related family (including KAT2A/B), MYST family (comprising TIP60, MOZ, MORF, HBO1, and MOF), and other proteins (consisting of P300/CREB-binding protein (CBP) complex and SRC group).

The upregulation of transcription factors is associated with epigenetic alterations, especially histone acetylation, in the ER signaling pathway. Upregulated levels of ERα and FOXA1, two well-known biological markers for ER-dependent breast cancer, are associated with the hyper-acetylation of active histone marks (H3K27ac, H3K9ac, and H3K14ac) at the enhancer [69,109,110]. HATs and associated proteins, such as P300, CBP, and P300/CBP associated factor (PCAF) are potential diagnostic biomarkers for ER+ breast cancer exhibiting H3K27ac enrichment. For example, P300 is a selective element that may enhance ERα activity in luminal breast cancer [111]. The upregulation of the P300/CBP complex regulates the expression of oncogenes and tumor suppressors, including those encoding c-MYC, AR, P53, and BRCA1, which are associated with poor prognosis and tumor recurrence in patients with breast cancer [112,113,114,115,116]. Additionally, ER signaling is regulated by MYST3 and histone acetylation in the MYST group. Thus, MYST3 can be a novel therapeutic target for luminal breast cancer [117]. Several studies have suggested that ERα regulates the expression of HBO1, while other HAT groups regulate luminal breast cancer proliferation and promote stemness of breast cancer stem cells [118,119].

Additionally, other histone acetylation modifications have been demonstrated to contribute to the development of ER+ breast cancer [103,120,121]. For example, H4K8ac mediates the transcription activity by broadening enhancers and crosstalk with the H3K27ac signal. Hence, histone modification is considered a potential diagnostic marker for luminal breast cancer [121]. The global upregulation of H3K4ac is associated with the signals for early transformation and aggressive metastasis in ER+ breast cancer [120]. Correspondingly, the inhibition of TIP60, which reduces the H3K4ac signal, can be a potential therapeutic strategy for ER+ breast cancer [122]. One study reported that the downregulation of H3K18ac is negatively correlated with the survival rate but not with tumor size, stage, or lymph node metastasis in patients with breast cancer [103]. Low or no H4K16ac signals are detected in most breast cancer samples [103,123,124,125].

HATs and HDACs are considered potential diagnostic markers for breast cancer tumorigenesis and endocrine therapy resistance as they regulate ER signals. HAT activity confers resistance to the cancer cells to hormone therapy through the regulation of the PI3K/Akt and MAPK pathway and downregulates the expression of E-cadherin, a member of the EMT signaling, after tamoxifen administration [126]. Upregulation of GCN5 (KAT2A) mitigates tamoxifen sensitivity by modulating P53 degradation [127]. The overexpression of the oncogene *AIB1* promotes breast cancer metastasis by regulating EMT signals, such as *PEA3*, *MMP2*, and *MMP9* [127,128,129]. Conversely, HDACs regulate the expression of ERα target genes by functioning as corepressors at the ERα-binding sites. Furthermore, some selective HDAC inhibitors, such as PCI, vorinostat, and panobinostat are reported to enhance the expression of apoptosis factors (P21, cMYC, and BCL2) in breast cancers [107]. ERα-mediated HDAC6 overexpression regulates microtubule expression and cell viability and migration [130,131]. HDAC6 is upregulated in more than 65% of ER+/PR+ breast tumors, which are associated with improved disease-free survival (DFS) but not overall survival (OS) [131,132]. The induction of HDAC6 may be correlated with high BCL2 levels and paclitaxel resistance in luminal breast cancer. Thus, alpha-tubulin or HDAC6 inhibitors can be potential therapeutic agents for luminal breast cancer [133,134]. Additionally, several HDAC members can serve as potential markers for ER+ breast cancer both in early breast cancer progression and endocrine therapy response [135]. Oncomine data analysis revealed that most HDACs, including *HDAC1*, *HDAC3*, *HDAC5*, *HDAC7*, *HDAC10*, and *HDAC11*, are upregulated in luminal breast cancer subtypes [135,136,137]. The upregulated levels of HDAC1 and HDAC3 are significantly associated with ERα and PR protein levels in 40% and 44% of breast cancer cases, respectively [138,139,140]. Patients with breast cancer exhibiting upregulated HDAC1 levels are associated with better DFS than OS [138,140].

#### 3.2.2. Histone Methylation and Demethylation

Upregulation of H3K4me3 levels is positively correlated with the survival rates of patients with breast cancer [136,141]. The H3K4me3 signal is preferentially enriched at the enhancer, promoter, and CTCF sites in endocrine therapy-resistant breast cancer cells [19,141].

KDM5, an H3K4me2/me3 demethylase in ER signaling, is involved in carcinogenesis and therapy resistance [142,143,144,145,146,147,148]. Hinohara et al. demonstrated that KDM5A and KDM5B are widely expressed in ER+ breast cancer and promote ER signaling-mediated downregulation of H3K4me3 levels using single-cell genomic approaches [143]. KDM5A and KDM5B, which are commonly overexpressed in luminal breast cancer, modulate ER signaling, alter gene expression profiles, and promote endocrine therapy resistance [143,146,149,150,151,152,153]. Additionally, KDM5A and KDM5B regulate cell proliferation and migration by promoting lipid metabolic reprogramming. Therefore, upregulated KDM5A and KDM5B levels are associated with poor clinical outcomes [150,152,153,154]. Yamamoto et al. revealed that KDM5B determines the luminal subtype and that KDM5B depletion can reverse the subtype of breast cancer from luminal to basal-like type [152]. KDM5C is a switching factor that balances the regulation of ER signaling and tamoxifen-induced sensitization in breast cancer cells [147]. Overexpression of KDM5C increases breast cancer cell proliferation and tumorigenesis by modulating the H3K4 methylation status and consequently promotes endocrine therapy resistance [145,147]. SMYD is a potential biomarker for ER+ breast cancer. This is because SMYD mediates ER signaling by demethylating H3K4 and/or H3K36. SMYD2 and SMYD3 directly interact with ERα to form a transcription complex and activate ERα target genes [155,156]. SMYD2 and SMYD3 are upregulated in 10% of patients with breast cancer and are significantly associated with cancer progression and poor prognosis in luminal breast cancer [155,156,157].

In the last decade, the members of the KMT2 (or mixed-lineage leukemia (MLL) family) and Complex Proteins Associated with Set1 (COMPASS) families have piqued the interest of the scientific community owing to their roles in regulating H3K4 and/or H3K36-methylated processes. A coregulator comprising the KMT2 family members and the SET1 family (COMPASS), plays a critical role in the transcriptional regulation in luminal breast cancer [158]. The KMT2 family includes the following eight members: KMT2A (MLL1), KMT2B (MLL2), KMT2C (MLL3), KMT2D (MLL4), KMT2E (MLL5), KMT2F (hSETD1A), KMT2G (hSETD1B), and KMT2H (ASH2). SET Domain Containing 1A (SETD1A), which is upregulated in luminal breast cancer, promotes breast cancer proliferation, migration, and metastasis [159]. Additionally, SETD1A upregulates the levels of matrix metalloproteinases (MMPs), including *MMP2*, *MMP9*, *MMP12*, *MMP13*, and *MMP17*, by demethylating H3K4me3 and consequently promotes cancer metastasis [159,160]. Chromatin complexes subunit BAP18, an H3K4me3 reader protein of the KMT2A-WDR5 complex, mediates ER signaling by recruiting COMPASS-like to EREs [161]. This study examined if the roles of KMT2A and its complex in ER+ breast cancer are similar to those of other KMT2 members. KMT2C and KMT2CD are the most common epigenetic factors harboring somatic mutations that promote breast tumorigenesis and therapy resistance [75]. In the luminal breast cancer subtype, KMT2C mutations (approximately 14%) occur at a higher frequency than KMT2D mutations (approximately 8.7%) based on the analysis of METABRIC and The Cancer Genome Atlas (TCGA) breast cancer datasets [75,162]. KMT2C promotes E2-stimulated ERα activity and breast cancer proliferation. The KMT2C and KMT2D levels are associated with survival outcomes after anti-estrogen treatment [163,164]. Recently, KMT2C and KMT2D were reported to physically interact with FOXA1 and ERα proteins and subsequently regulate the activation of ERα target genes [165,166]. E2 induction promotes the enrichment of KMT2C and KMT2D at H3K4me1-marked enhancers and the signals may be unique for the proliferation of breast cancer cells [75,163,167]. The high prevalence of KMT2C mutations changes the landscape of H3K4me1 enrichment and ERα binding in luminal breast cancer [163,165]. High frequencies of KMT2C and KMT2D mutations also downregulate ERα and FOXA1 expression, suggesting that the loss of KMT2C/D can reduce the sensitivity of luminal breast cancer to ER inhibitors by activating the hormone-independent pathway [163]. Therefore, aberrant expression of KMT2C/D and SETD1A is a potential biomarker for luminal type and endocrine-resistant breast cancer [159,160,164,165,166,168].

H3K9 methylation is a marker of transcription silencing, which is closely correlated with ER signaling in breast carcinogenesis [169,170,171]. KDM4A (known as JMJD2A or JHDM3), an H3K9 and H3K36 demethylase, is upregulated in most breast samples. The expression of KDM4A is correlated with pathological parameters, such as tumor grade, histology, and survival rate [172,173]. KDM4A constitutes a part of the hormone complex that stimulates ER signaling and cancer cell growth [172]. KDM4B (JMJD2B/TDRD14B) may have an integral function in forming a complex with MLL2, which methylates histone H3 at both K4 and K9 sites [174]. E2 induction regulates the expression of KDM4B, which promotes ER+ breast cancer-associated gene expression [174,175]. Similar to KDM4A, KDM4B upregulation facilitates ER signaling. KDM4B, which is recruited to the transcription machinery of ERα target genes, regulates estrogen-induced tumor cell proliferation [174,176,177].

Single-cell visualization technologies, which are powerful approaches in cancer biology research, allow the exploration of a small number of tumor cell subpopulations that may contribute to cancer recurrence after treatment. Single-cell chromatin immunoprecipitation-sequencing analysis revealed that a subset of cells with low H3K27me3 signals in untreated tumors is the main cause of hormone therapy resistance in breast cancer [178]. Loss of the H3K27me3 signal is associated with the upregulation of several breast cancer oncogenes, such as those encoding epidermal growth factor receptor (EGFR) or insulin like growth factor binding protein 3 (IGFBP3) involved in tamoxifen resistance [178,179]. The low level of H3K27me3 is likely due to EZH2 (KMT6A, ENX1) upregulation, a common feature of the ER-negative breast cancer subtype. However, EZH2 upregulation is also a risk factor in ER-positive breast cancer as it is associated with the formation of the ERα-β-catenin complex [180,181]. As alterations in EZH2 expression alone are insufficient for breast cancer development, additional factors along with estrogen-induced EZH2 overexpression are necessary for ER+ breast cancer development [182,183]. The crosstalk between ER signaling and Wnt signaling is mediated by the polycomb complex protein EZH2. This indicates that EZH2 promotes the cell cycle and increases tumor cell proliferation [63,184,185]. Additionally, overexpression of MEL-18, a component of the polycomb complex, is reported to induce ERα activity in luminal breast cancer [64]. The downregulation of MEL-18 expression leads to the suppression of luminal breast cancer markers, whereas the overexpression of MEL-18 activates ER signaling. These findings indicate that MEL18 is a potential anti-hormonal therapeutic agent.

The function of KDM6A (UTX), an H3K27me3-demethylase, is associated with luminal breast cancer cell proliferation and metastasis [164,186]. Similar to KDM4B, KDM6A is also induced by ERα and forms an ERα transcription activator in breast cancer during hormone stimulation [186]. KDM6A physically interacts with ERα upon E2 treatment, demethylates H3K27me3 to facilitate the expression of C-X-C Motif Chemokine Receptor 4 (CXCR4) oncogene or the pluripotency factors NANOG, SOX2, and KLF4, which are related to breast cancer metastasis [186,187]. OCT4, which is frequently upregulated in breast cancer [188], regulates breast cancer cell survival by modulating H3K27me3 and H3K27ac histone marks and regulating a subset of genes related to cell proliferation and metastasis [187,188]. EMT signals are also affected by KDM6A-mediated and KDM2D-mediated changes in H3K27me3 and H3K4me3 levels [164,189]. KDM6A mutation leads to the formation of complexes with the PRC co-repressor, where EZH2 is activated instead of forming complexes with the UTX co-activator [190,191]. Svotelis et al., reported that the KDM6B-ERα complex directly regulates apoptotic programs by H3K27 demethylation-mediated activation of BCL2, which promotes resistance to endocrine therapies in luminal breast cancer [192]. KDM6B, which is upregulated in luminal breast cancer, was significantly associated with the upregulation of EMT signals and poor clinical prognosis [193,194]. Enhanced levels of KDM6B are also correlated with EZH2 dysfunction in EREs and resistance to PI3K/AKT inhibitor treatment [192,194]. Therefore, the co-regulatory circuit of ERα and H3K27 demethylase could be a potential biomarker for luminal breast tumorigenesis and metastasis.

## 4. Chromatin Remodelers and ER+ Breast Cancer

More than 30% of breast cancers are associated with epigenetic reprogramming, which activates the EMT signaling [84,195]. Dysfunction of chromatin remodelers, which is reported to promote early tumorigenesis in the breast, is correlated with clonal diversity and genomic instability in breast cancer [55,196,197,198,199]. The studies suggest that modulation of chromatin dynamics is an optional approach to ameliorate subtypes of breast cancer.

The SWI/SNF complex promotes breast cancer. ARID1A is frequently downregulated or mutated in approximately two of three breast cancer cases [198,199]. A high frequency of ARID1A mutations was detected in luminal breast cancer with a tumor suppressor gene score of 45% [190]. Loss of ARID1A inhibits the ER antagonist response and promotes breast cancer recurrence after anti-estrogen therapy [196,197]. ARID1A dysfunction is associated with impairment of the BAF complex and altered HDAC1/BRD4-driven transcription activities to suppress tumor growth [196]. Furthermore, Xu et al. demonstrated that the downregulation of ARID1A levels causes loss of co-occupancy of ERα and FOXA1 at the SWI/SNF-binding sites and induces cellular plasticity from a luminal-like state to a basal-like state [197]. BRG-1, an SWI/SNF complex member, is recruited to ERα-responsive regions in response to E2 stimulation and alters the histone acetyltransferase activity of the CBP/P300/PCAF complex in luminal breast cancer [55].

The histone variant H2A.Z and its eviction factors (INO80 complex) are reported to regulate the expression of ER-dependent breast tumor markers, such as *TFF1* and *GREB1* [50,200]. Low levels of H2A.Z induce EMT signaling during breast cancer progression by activating the TGF-β pathway [201].

## 5. Epi-Drugs for Breast Cancer

Although hormonal therapy is highly efficacious, alternative therapeutic approaches for ER+ breast cancer subtypes must be explored as the cancer cells exhibit endocrine therapy resistance and epigenomic reprogramming. In contrast to genetic alterations, epigenetic alterations are dynamic and reversible. Therefore, current hormonal therapy can be further improved to overcome therapy resistance. Premenopausal patients with luminal breast cancer are eligible for endocrine therapy, which markedly decreases tumor size. Tamoxifen and fulvestrant, which are well-known anti-estrogen agents, exhibit potent antitumor efficacy in premenopausal patients. In the post-menopausal stage, E2 production is regulated by aromatase in the extra-ovarian tissues instead of the ovary in which E2 is no longer synthesized [202]. The efficacy of AIs is higher than that of ER antagonists in post-menopausal patients with luminal breast cancer. Letrozole, anastrozole, and exemestane, which are AIs that competitively bind to aromatase, have been successfully used for the clinical treatment of hormone-positive and HER2-negative breast cancers [203]. Epi-drugs alone or in combination with endocrine drugs have been considered for ER+ breast cancer treatment to mitigate endocrine therapy resistance.

Therapeutic approaches other than hormonal therapies should be considered to treat luminal breast cancer harboring mutations in epigenetic factors. For example, breast tumors with ARID1A mutations exhibit decreased genome-wide binding of ERα and FOXA1, which leads to the loss of luminal cell characteristics and promotes endocrine therapy resistance [196,197]. As ARID1A forms a complex with HDAC1, ARID1A mutation leads to the loss of HDAC1 function and promotes the progression of luminal breast cancer in a BRD4-dependent manner. Bromodomain and extra-terminal motif (BET) inhibitors, including JQ1 and IBET762, exerted significant growth-inhibitory effects against ARID1A-null breast cancer (Figure 2 and Table 1). This suggests that the combination of BET inhibitor and hormonal therapy is potentially effective against luminal breast cancer harboring mutant ARID1A [196]. Other studies have demonstrated that KMT2C is an epigenetic factor with a high mutation frequency in hormonal receptor-positive and HER2 negative metastatic breast cancers [75,162]. The loss of KMT2C promotes breast cancer cell reprogramming by downregulating H3K4me1 and upregulating FOXA1-AP-1 binding [163]. ER antagonists (including fulvestrant, AZD9496, ARN1917, GDC927, and RU58668) and tamoxifen can sensitize KMT2C-depleted luminal breast cancer [163,204]. Some studies suggest that luminal breast cancer harboring KMT2C mutation can be treated with the EZH2 inhibitor GSK126, which mitigates dysregulated gene expression patterns and aberrant cell proliferation [158,168]. Therefore, these studies suggest that combinatorial therapy with anti-EZH2 and anti-hormone agents is a potential therapeutic approach for ER+ breast cancer cells harboring mutant KMT2C (Table 1).

Targeting histone acetylation is a potential therapeutic strategy for breast cancer as it may resensitize HATs and HDACs associated with ER signaling. HAT inhibitors are epi-drugs that downregulate histone acetylation levels. Some promising HAT inhibitors serve as epi-drugs, such as TIP60-specific inhibitors (TH1834, garcinol, pentamidine, NU9056, and bisubstrate inhibitor A) and multi-HAT inhibitor (MG-149, curcumin, Lys-CoA, and anacardic acid) for the treatment of breast cancer (Table 1). TH1834 exerts growth-inhibitory effects against breast cancer by specifically inhibiting TIP60, which inhibits the response of cancer cells to DNA repair [205]. Additionally, TH1834 decreased the tumor size in a xenograft tumor model by targeting the TIP60-ER signaling pathway [206]. Garcinol mitigates E2-induced cell proliferation and promotes cell cycle arrest and apoptosis in MCF7 cells by downregulating the cyclin D1, BCL-2, and BCL-XL levels [207].

HDAC inhibitors are categorized into the following four subgroups based on their chemical structure and range of activity: hydroxamates, cyclic peptides, aliphatic acids, and benzamides. As HDACs are critical for ER-mediated transcription regulation, several HDAC inhibitors have been used in long-term preclinical and clinical therapies for ER+ breast cancer (Figure 2 and Table 1).

Entinostat, a class I HDAC (HDAC1 and HDAC3) inhibitor, upregulates the expression of ERα in ER-negative and ER-positive breast cancers and enhances the sensitivity of cancer cells to endocrine therapy, especially in ER-silent metastatic breast cancer [208,209,210]. Furthermore, entinostat exerts immunomodulatory effects and downregulates the expression of CD40 in granulocytes and monocytes and upregulates the expression of Human Leukocyte Antigen-DR isotype (HLA-DR) on cluster of differentiation (CD) 14+ monocyte surfaces. This leads to the potentiation of antitumor activity by upregulating cytotoxic T-lymphocyte-associated protein 4 (CTLA-4) and Programmed cell death protein 1 (PD-1/CD279) expression in T-cells [211]. Entinostat has been evaluated in at least 20 clinical trials for the treatment of breast cancer. The combination of entinostat and exemestane significantly improved the median progression-free survival (4.3 months vs. 2.3 months) and OS (28.1 months vs. 19.8 months) in patients with post-menopausal advanced ER+ breast cancer [212]. Additionally, the combination of entinostat and exemestane was evaluated in several clinical trials on patients with post-menopausal ER+ breast cancer (NCT01594398, NCT02820961, NCT02115282, NCT03538171, NCT02833155, and NCT00828854). The combination of entinostat and fulvestrant or tamoxifen (ENCORE305, NCT02115594) enhanced the efficacy of endocrine therapy in breast cancer and metastatic breast cancer [107].

Several promising HDACis, including abexinostat (PCI-24781), panobinostat (LBH589), and vorinostat (suberoylanilide hydroxamic acid [SAHA]), are available for the treatment of ER+ breast cancer. Vorinostat increased the efficacy of anti-estrogen therapy [107,213]. In breast cancer and tamoxifen-resistant cell line models, vorinostat treatment upregulates the expression of autophagy-mediated cell death markers, such as *LC3-II* and *beclin-1*. Vorinostat yielded positive therapeutic outcomes in phase II clinical trials (NCT00365599 and NCT01194427) in patients for whom tamoxifen or AI therapy failed (NCT01720602). HDAC inhibitors, such as sodium butyrate, panobinostat, and vorinostat, which can impair DNA repair, suppress the escape of MCF7 cells from DNA damage response and cell death program and exert cytotoxic effects on metastatic breast cancer cells by inhibiting the EMT signaling pathway [214,215,216]. WT161, an HDAC6 inhibitor, can sensitize luminal breast cancer more effectively than other subtypes by activating the apoptotic protein X-Linked Inhibitor of Apoptosis (XIAP) and downregulating EGFR, HER2, and ERα [217]. Thus, HAT and HDAC inhibitors can serve as potential epi-drugs for the treatment of ER-dependent breast cancer.

Based on the biomarkers for luminal breast cancer, dysfunctional DNMT alters the balance between genetic and involved factors. Current studies have majorly focused on therapeutics regulating DNA methylation for the treatment of luminal breast cancer (Figure 2 and Table 1) [96,97]. The Food and Drug Administration-approved DNA hypomethylating compound 5-aza-2′deoxycytidine (also called decitabine, 5-azadC, 5-azacytidine), which re-sensitizes breast cancer, is an alternative to endocrine therapy. In MCF7 cells, 5-azadC inhibits DNMT3B activity and activates Apoptotic Peptidase Activating Factor 1 (APAF1), an apoptosis marker [218]. Additionally, 5-azadC suppresses the expression of *DNMT1*, *IGF4*, and *ERRα* (*NR3B1*) and mitigates breast cancer progression [219]. Furthermore, 5-azadC upregulates the levels of *SALL2* in luminal breast cancer and consequently increases the sensitivity of cancer cells to tamoxifen. Hence, the coadministration of tamoxifen and DNMT inhibitors may provide beneficial therapeutic outcomes for patients with breast cancer [220]. The combination of 5-azacytidine and the HDAC inhibitor trichostatin A was previously investigated as a therapeutic for endocrine therapy-resistant luminal breast cancer. This combination promoted hypermethylation at the promoters of the genes encoding *NR2F2*, *LCC2*, and *LCC9* and increased the sensitivity of cancer cells to anti-estrogen therapy [221]. RG108, a DNA methylation inhibitor, downregulated the expression of ER coregulators, including NTRK2, NR2F2, CTDP1, SETBP1, and POU3F2. Thus, the co-administration of RG108 and tamoxifen increases the sensitivity of MCF7 cells to endocrine therapy [222]. Proof-of-concept studies on DNA methylation inhibitors further support that methylation plays a critical role in increasing the efficacy of endocrine therapies by mitigating drug resistance.

Regulating global histone methylation is a promising strategy for suppressing or activating transcription in luminal breast cancer cells (Figure 2 and Table 1). The inhibitors of KDM1A, an enzyme responsible for H3K4 methylation, suppress aromatase expression and increase the sensitivity of cancer cells to hormone therapy [223,224]. Therefore, combinatorial therapy with drugs targeting KDM1 and endocrine therapies, such as tamoxifen and letrozole can be an optimal treatment strategy for luminal breast cancer. The combination of tamoxifen and pharmacological KDM1 inhibitors, such as pargyline and NCL1 [N-((1S)-3-(3-(trans-2-aminocyclopropyl) phenoxy)-1-(benzylcarbamoyl) propyl) benzamide] is reported to significantly decrease breast cancer proliferation and tumor size [225]. MC3324 can alter the function of KDM1A and KDM6A, which leads to significant changes in H3K4 and H3K27 trimethylation levels, respectively. In addition to its function in histone modification, M3324 arrests cell growth and induces apoptosis in luminal breast cancer and resistant models. Oral administration of MC3324 efficiently repressed tumor growth in a breast cancer xenograft model in combination with an anti-estrogen agent [226]. Administration of KDM5 inhibitors can be a novel therapeutic strategy for ER+ breast cancer as they decrease the pre-existing endocrine therapy-resistant cell populations. Previous studies have demonstrated that a small-molecule inhibitor of the KDM5 family enzymes KDM5-C49 (C49) and its cell-permeable ethyl ester derivative (KDM5-C70) increased the response of luminal breast cancer cells to hormonal therapy [143,152,227,228]. DOT1L inhibition is sufficient to cause growth arrest in luminal breast cancer cells, which suggested that H3K79 methylation has functional significance in ER signaling. The DOT1L inhibitor EPZ004777 downregulated ERα levels and markedly decreased tumor size after 3 weeks of treatment in hormone-positive and anti-estrogen-resistant breast cancer cells [229]. Therefore, the identification of promising drugs is important to validate the role of histone modification in luminal breast cancer and develop potential novel therapies.

**Table 1 biomedicines-10-00748-t001:** Epi-drugs examined for luminal breast cancer treatment.

Chemicals	The Target of Epigenetic Factor	Co-Administrated with	Mechanism	Stage of Study	Condition	Reference
TH1834	TIP60		Prevents cancer cell response DSB repair capacity	Preclinical	MCF7 cell line and Xenograft	[206]
Garcinol	Acetyltransferase inhibitor		Decreases ac-p65 protein expression level in the NF-κB pathway	Preclinical	MCF7	[207]
Entinostat	HDAC1 and HDAC3 inhibitor	Exemestane	Induces apoptosis by reversing BCL-2 overexpression	Phase II NCT02820961	ER (+) breast cancer	[107]
Vorinostat	HDAC2 inhibitor	Tamoxifen AIs (anastrozole, letrazole, exemestane)	Reduces HDAC2 level and increasing hormonal therapy	Phase II NCT00365599 NCT01194427 Phase I NCT01720602	ER (+) breast cancer	[213]
Vorinostat	HDAC class I	Ionizing radiation	Enhances DNA damage through the inhibition of DNA repair	Preclinical	MCF7	[215]
WT161	HDAC6 inhibitor		Downregulates apoptosis protein XIAP and luminal breast cancer marker EGFR, HER2 and ERα	Preclinical	ER (+) breast cancer	[217]
Sodium butyrate	Inhibit the H4 deacetylation	Etoposide	Reduces DSB repair capacity	Preclinical	MCF7	[214]
5-azacytidine	DNMTs inhibitor	Tamoxifen	Reactivates APAF-1 and SALL2 to re-sensitized with anti-estrogen therapy	Preclinical	MCF7	[218,220]
RG108	DNMTs inhibitor	Tamoxifen	Reactivates SRC-1 to expression ER-coregulator	Preclinical	ER (+) breast cancer	[222]
NCL-1 benzamide or pargyline	KDM1s inhibitor	Tamoxifen	Modifies histone marks at ERα target gene promoter	Preclinical	ER (+) breast cancer	[225]
MC3324	Dual function in KDM1A and KDM6A inhibitor		Reduces ERα levels and suppression ER-coactivator as well as increase H3K4me2 and H3K27me3	Preclinical	ER (+) breast cancer	[226]
JQ1 or IBET762	BET inhibitor	Fulvestrant Tamoxifen	Inhibits BRD4 overexpression in ARID1A-deficient cell	Preclinical	ER (+) and ARID1A-deficient breast cancer cell	[196]
KDM5-C49 or KDM5-C70	KDM5A/B inhibitor	Fulvestrant	Targets KDM5 family that cause H3K4me3 reduction	Preclinical	MCF7, T47D	[143]
GSK126	EZH2 inhibitor		Inhibits H3K27 methylation	Preclinical	ER (+) breast cancer cell harboring KMT2C mutation	[158]
EPZ5676	DOT1L inhibitor	Fulvestrant Tamoxifen	Reduces H3K79 methylation levels and blocks ERα signaling	Preclinical	MCF7, T47D, TAM-resistant and ICI-resistant ER (+) breast cancer cell	[229]

## 6. Perspectives

Rapid progression has been achieved in elucidating the function of ERα in the regulation of breast cancer pathogenesis. However, the epigenetic mechanism underlying the process of hormone response and hormone therapy resistance in breast cancer cells must be further elucidated. Although endocrine therapies have yielded beneficial outcomes in patients with luminal breast cancer, some breast cancer cells exhibit resistance to endocrine therapy by activating several pathways to escape the cell death program. Epigenetic modifications define the stages of breast cancer development. Recent studies have elucidated the epigenetic and genetic molecular mechanisms, especially ER signaling, in luminal breast cancer, which have contributed to improving treatment outcomes. Rapid advances in NGS technology have enabled the classification of luminal breast cancers based on four classical biomarkers (ER, PR, HER2, and KI67) and the identification of various molecular genetic and epigenetic defects. Therefore, profiling molecular features in individual patients will enable the development of improved management strategies for breast cancer.

Epi-drugs, which were recently reported to be potential therapeutic agents for cancer, can mitigate the recurrence or metastasis of luminal breast cancer. For example, several epi-drug candidates, such as HDAC or histone modification inhibitors were reported to suppress ER signaling and breast cancer growth. Additionally, epi-drugs can potentiate the efficacy of hormone therapy. Furthermore, some phase I and phase II clinical trials have reported that epi-drugs improve overall survival outcomes in patients with luminal breast cancer. Preliminary studies suggest that epigenome editing using the clustered regularly interspaced palindrome repeat/caspase 9 technique can be an alternative therapeutic strategy for breast cancer. However, further studies are needed to investigate the functional involvement of epigenetic factors in luminal breast cancer to elucidate the underlying mechanism and identify potential diagnostic and therapeutic biomarkers. In conclusion, a single therapeutic does not exert optimal therapeutic effects on ER-positive breast cancer. Thus, the therapeutic strategies for ER-positive breast cancer must be further optimized and diversified. The co-administration of epi-drugs and anti-cancer agents can be a novel approach for developing personalized medicine.

## Figures and Tables

**Figure 1 biomedicines-10-00748-f001:**
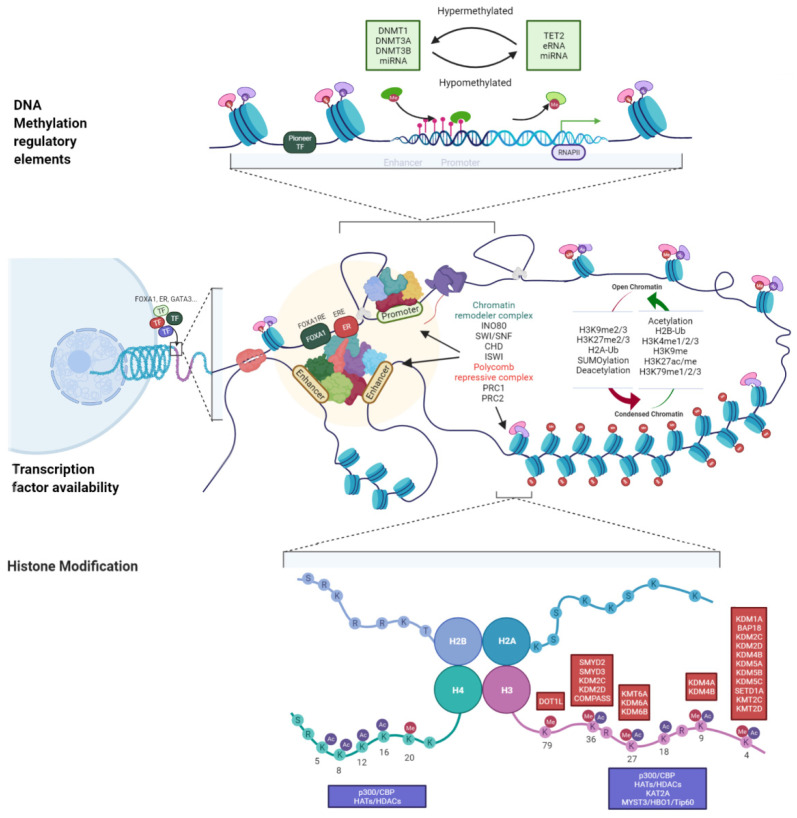
Epigenetic regulation in estrogen receptor signaling. Estrogen receptor-alpha (ERα)-related epigenetic changes in luminal breast cancer. Aberrant regulation of ERα alters the function/expression of coregulators, pioneer transcription factors (such as FOXA1), and epigenetic factors regulating DNA methylation, histone modification, and non-coding RNAs in luminal breast cancer. TF, transcription factor; ERE, estrogen response element; FOXA1RE, FOXA1 regulatory element; RNAPII, RNA polymerase II; S, Serine; R, Arginine; K, Lysine; T, Threonine; Me, methylation; Ac, acetylation; H, histone. The figure is created by the BioRender tool.

**Figure 2 biomedicines-10-00748-f002:**
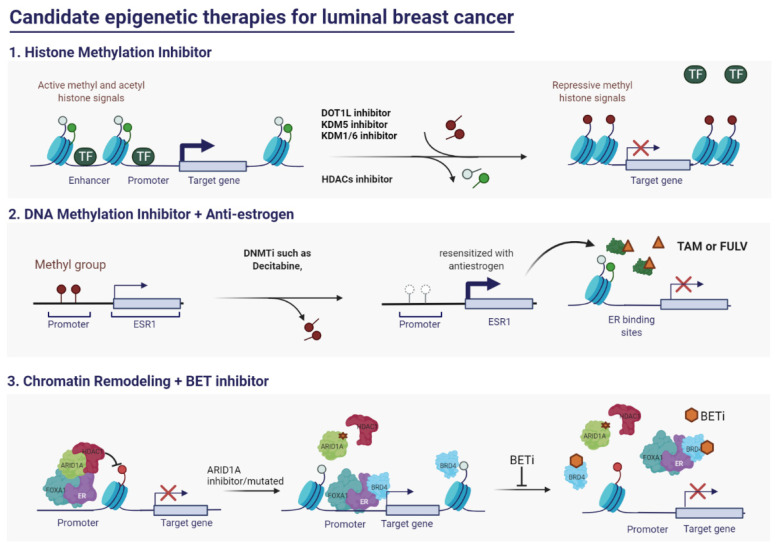
Epi-drugs used in luminal breast cancer therapy: (**1**). Histone methylation inhibitors exhibit growth-inhibitory effects against luminal breast cancer by suppressing estrogen receptor (ER)-alpha (ERα) function and expression. DOT1L (H3K79 methylation), KDM5/KDM1 (H3K4 demethylation), and KDM6 (H3K4 and H3K27 demethylation) inhibitors have been evaluated as potential therapeutics for ER-positive breast cancer. (**2**). The sensitivity of luminal breast cancer with downregulated ER expression to anti-estrogen therapy can be potentiated by co-treatment with DNA methylation inhibitors. (**3**). Mutations in the chromatin remodeler lead to alterations in chromatin landscape and gene expression in luminal breast cancer. For example, ARID1A mutation switches HDAC1-mediated gene suppression to BRD4-mediated gene activation. Therefore, BET inhibitors can be effective against ARID1A-deficient luminal breast cancer. TF, transcription factor; DNMTi, DNMT inhibitor; TAM, tamoxifen; FULV, fulvestrant; BET, Bromodomain and Extra-Terminal motif. The figure is created by the BioRender tool.

## Data Availability

Not applicable.

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
