# Peer review of "Epigenetic Factors as Etiological Agents, Diagnostic Markers, and Therapeutic Targets for Luminal Breast Cancer"

_biomedicines, 2022, doi:10.3390/biomedicines10040748_

Round 1

Reviewer 1 Report

In the manuscript the authors present a review on the involvement of epigenetic changes in ER+ breast cancer onset and progression. They summarized recent findings on epigenetic and transcription factors as new potential cancer biomarkers and therapeutic targets for tailored medicine. the recent use of the main epi-drugs was also reported. The paper is well-organized and encourage further studies to better clarify the role of epigenetic factors in luminal breast cancer development. However, I have some minor comments on this article as follows:

- Page 1 line 34: The authors report the percentage of new breast cancer cases. However, it is not indicated if the number is referred to both sexes or not. Please specify.

- Page 1 lines 43-45: The authors report that ERα upregulation characterizes 75-80% of all breast cancer cases. The sentence should be supported by a reference.

- Page 3 lines 94-97: The sentence “ER signaling is critical for physiological mammary gland development and pathological tumorigenesis. The healthy breast tissue is reported to comprise < 20% ERα-positive cells. Hence, enhanced ER activity and ER-positive cell number are high-risk factors for breast carcinogenesis” needs a supporting reference.

- Page 4 line 128: The sentence “Histone acetyltransferases (HATs) may function act as…..” must be corrected choosing “may function” or “may act”.

- Page 6 lines 256-258: The paragraph entitled “3.2. Aberrant histone modifications in ER+ breast cancer” is too much short. The authors must integrate the section with other supporting information.

- Page 10 lines 430-432: The authors should add a reference to the sentence “Dysfunction of chromatin remodelers, which is reported to promote early tumorigenesis in the breast, is correlated with clonal diversity and genomic instability in breast cancer”.

- Page 11 lines 480-482: The sentence “Some studies suggest that luminal breast cancer harboring KMT2C can be treated with the EZH2 inhibitor GSK126, which mitigates dysregulated gene expression patterns and aberrant cell proliferation” is supported by just one reference. Since the authors indicate “some studies”, other references must be added.

- Page 12 lines 569-572: Only one reference supports the sentence “Previous studies have demonstrated that a small-molecule inhibitor of the KDM5 family enzymes KDM5-C49 (C49) and its cell-permeable ethyl ester derivative (KDM5-C70) increased the response of luminal breast cancer cells to hormonal therapy”. Please add other references.

- The extended full name of genes, proteins, transcription factors, and co-regulators must be reported when they are cited for the first time along the text (e.g. lines 46, 106, 131, 138, 139,167, 169, 226, 229, 235-237, 251, 264-266, 279, 281, 301, 338, 349-350, 357-358, 394-395, 413, 433, 438,541, 547, 550 etc).

- Full gene names and abbreviations must be italicized (e.g. lines 46, 226, 251, 277, 300, 371, 467 etc).

- Figure 1 and Figure 2 are not reported along the text. Both figures must be mentioned in the corresponding section.

- Table 1 must be mentioned along the text in the corresponding paragraph.

Author Response

The authors thank the reviewer very much for his/her valuable comments and support. The following is our point-by-point responses for the comments.

- Page 1 line 34: The authors report the percentage of new breast cancer cases. However, it is not indicated if the number is referred to both sexes or not. Please specify.

Response: We have added "both male and female" at line 35 in page 1.

- Page 1 lines 43-45: The authors report that ERα upregulation characterizes 75-80% of all breast cancer cases. The sentence should be supported by a reference.

Response: We have added the following references; PMID6491696, PMID10963602 and PMID15535853

- Page 3 lines 94-97: The sentence "ER signaling is critical for physiological mammary gland development and pathological tumorigenesis. The healthy breast tissue is reported to comprise < 20% ERα-positive cells. Hence, enhanced ER activity and ER-positive cell number are high-risk factors for breast carcinogenesis" needs a supporting reference.

Response: We have added the following references; PMID10819502 and PMID3664479.

- Page 4 line 128: The sentence "Histone acetyltransferases (HATs) may function act as….." must be corrected choosing "may function" or "may act".

Response: We have changed it to "may function" at line 129 in page 4.

- Page 6 lines 256-258: The paragraph entitled "3.2. Aberrant histone modifications in ER+ breast cancer" is too much short. The authors must integrate the section with other supporting information.

Response: We have integrated the sentence into 3.2.1.

- Page 10 lines 430-432: The authors should add a reference to the sentence "Dysfunction of chromatin remodelers, which is reported to promote early tumorigenesis in the breast, is correlated with clonal diversity and genomic instability in breast cancer".

Response: We have added the following references; PMID11003650, PMID31913353, PMID31932695, PMID21889920, PMID24430365.

- Page 11 lines 480-482: The sentence "Some studies suggest that luminal breast cancer harboring KMT2C mutation can be treated with the EZH2 inhibitor GSK126, which mitigates dysregulated gene expression patterns and aberrant cell proliferation" is supported by just one reference. Since the authors indicate "some studies", other references must be added.

Response: We have added one more reference; PMID31897900

- Page 12 lines 569-572: Only one reference supports the sentence "Previous studies have demonstrated that a small-molecule inhibitor of the KDM5 family enzymes KDM5-C49 (C49) and its cell-permeable ethyl ester derivative (KDM5-C70) increased the response of luminal breast cancer cells to hormonal therapy". Please add other references.

Response: We have added the following references; PMID27427228, PMID27214403 and PMID24937458

- The extended full name of genes, proteins, transcription factors, and co-regulators must be reported when they are cited for the first time along the text (e.g. lines 46, 106, 131, 138, 139,167, 169, 226, 229, 235-237, 251, 264-266, 279, 281, 301, 338, 349-350, 357-358, 394-395, 413, 433, 438,541, 547, 550 etc). Full gene names and abbreviations must be italicized (e.g. lines 46, 226, 251, 277, 300, 371, 467 etc).

Response: We have provided full name of genes in the revised manuscript.

- Figure 1 and Figure 2 are not reported along the text. Both figures must be mentioned in the corresponding section. Table 1 must be mentioned along the text in the corresponding paragraph

Response: We have added the following words to mention figures and table in the revised manuscript;

 "As shown in the figure 1," in page 4 line 123

"Figure 1" in line 105 page 3; 120 page 4

“Figure 2 and Table 1” in line 439 and 467, page 10; line 503-504 and line 524, page 11

"Table 1" in line 451 and 457, page 10

Reviewer 2 Report

After carefully read the review article:” Epigenetic factors as etiological agents, diagnostic markers, and 2 therapeutic targets for luminal breast cancer” I suggest a revision even if the authors have thoroughly investigated the subject.  While the introduction section seems to contain superficial information, the other paragraphs are full of information without following a logical thread. Inthe scientific literature there are several articles about breast cancer research  but in this review they are only listed and should be commented on. Finally, I suggest a total revision of the text in order to provide a good overview of the hot topic.

Author Response

The authors thank the reviewer very much for his/her insightful comments.

In the revision, we have removed sentences and paragraphs describing non-coding RNAs in page 4, 5 and 6 as the epi-drugs in the figure2 and table1 are mainly related to control of histone modifications, DNA methylation and chromatin remodelers.

We have also added some comments (line265-267, line449-451) in the revision.   

Reviewer 3 Report

In my oppinion no revisions are needed and I support accepting this review for publication in Biomedicines. Well written and very extensive review with overwhelming number of refferences. The structure is clear.

Author Response

The authors thank the reviewer so much for his/her support.

Round 2

Reviewer 2 Report

The authors reworked the text